# *RET*-Altered Cancers—A Tumor-Agnostic Review of Biology, Diagnosis and Targeted Therapy Activity

**DOI:** 10.3390/cancers15164146

**Published:** 2023-08-17

**Authors:** Antoine Desilets, Matteo Repetto, Soo-Ryum Yang, Eric J. Sherman, Alexander Drilon

**Affiliations:** 1Memorial Sloan Kettering Cancer Center, New York, NY 10065, USA; desilea@mskcc.org (A.D.); repettom@mskcc.org (M.R.); yangs2@mskcc.org (S.-R.Y.); shermane@mskcc.org (E.J.S.); 2Department of Oncology and Hemato-Oncology, University of Milan, 20133 Milan, Italy; 3Department of Medicine, Weill Cornell Medical College, New York, NY 10065, USA

**Keywords:** *RET* fusion, *RET* mutation, selpercatinib, pralsetinib, tumor agnostic

## Abstract

**Simple Summary:**

Changes in the *RET* gene (like mutations or fusions) are often found in lung and thyroid cancers but are also found in other cancer types. New drugs called “selective RET inhibitors”, like selpercatinib and pralsetinib, are effective in treating tumors with *RET* gene changes. These drugs have been tested in “basket trials” that treat patients based on gene changes in their cancer instead of cancer type. In this review, we discuss how *RET* gene changes cause cancer, which cancers have these changes, what tests to use, and how well targeted therapies work.

**Abstract:**

*RET* alterations, such as fusions or mutations, drive the growth of multiple tumor types. These alterations are found in canonical (lung and thyroid) and non-canonical (e.g., gastrointestinal, breast, gynecological, genitourinary, histiocytic) cancers. *RET* alterations are best identified via comprehensive next-generation sequencing, preferably with DNA and RNA interrogation for fusions. Targeted therapies for *RET*-dependent cancers have evolved from older multikinase inhibitors to selective inhibitors of RET such as selpercatinib and pralsetinib. Prospective basket trials and retrospective reports have demonstrated the activity of these drugs in a wide variety of *RET*-altered cancers, notably those with *RET* fusions. This paved the way for the first tumor-agnostic selective RET inhibitor US FDA approval in 2022. Acquired resistance to RET kinase inhibitors can take the form of acquired resistance mutations (e.g., RET G810X) or bypass alterations.

## 1. Introduction

Advances in molecular profiling and inclusive trial designs have transformed the therapeutic landscape for patients with cancer. Basket trials now enroll patients based primarily on the presence of oncogenic drivers, such as *RET* (rearranged during transfection) alterations, regardless of tumor type. The landmark 2022 US FDA regulatory approval of selpercatinib, a RET selective tyrosine kinase inhibitor (TKI), represented the seventh tissue-agnostic indication in oncology [1], following tumor site-specific approvals of selpercatinib for non-small cell lung cancer (NSCLC) and thyroid cancer. *RET* alterations encompass both *RET* fusions and *RET* point mutations. Although NSCLC, papillary thyroid cancer (PTC), and medullary thyroid cancer (MTC) collectively account for the majority of *RET*-altered cancers, a plurality of additional tumor histologies also harbor *RET* alterations. In this review, we discuss the biology of *RET* alterations and strategies to target these oncogenic drivers, including contemporaneous tumor agnostic approaches based on registrational data from the selpercatinib and pralsetinib trials.

## 2. RET Biology

The *RET* proto-oncogene is a receptor tyrosine kinase (RTK) which functions as a transmembrane glycoprotein. The *RET* gene resides on the long arm of chromosome 10 and contains three functional domains: an extracellular ligand-binding domain, a transmembrane domain, and an intracellular tyrosine kinase domain [2]. RET is involved in normal embryonic development of the genitourinary, neuroendocrine, and sympathetic nervous systems [3,4]. Inactivating germline *RET* mutations and *Ret* knock-out mice models have yielded valuable insights into the consequences of reduced or absent RET activity throughout development and adulthood. Inactivating *RET* mutations have been identified in approximately 50% of familial cases of Hirschsprung disease, a condition characterized by the absence of enteric ganglion cells in the large intestine due to failure of neural crest cells to migrate completely during embryological development [5,6]. *RET* mutations are also associated with congenital abnormalities of the kidney and urinary tract (CAKUT) and congenital central hypoventilation syndrome (CCHS) [7,8]. In such disease entities, genotype–phenotype correlation remains poor, with variable penetrance and a wide range of *RET* mutations across the entire length of the gene [5]. Gain-of-function alterations are also observed; these are most commonly associated with oncogenesis and will be discussed in greater detail later.

Unlike many other RTK ligands, RET ligands such as artemin (ARTN), glial cell line-derived neurotrophic factor (GDNF), neurturin (NRTN), and persephin (PSPN) exhibit a distinctive binding mechanism [9,10,11]. These GDNF family ligands (GFLs) bind co-receptors known as GDNF family receptors (GFRs) [12,13]. Concurrent with binding of calcium ions to RET’s cadherin-like domain, the GFL–GFR complex induces recruitment of RET through its extracellular domain, thus bringing two RET monomers in close proximity. Receptor homodimerization then promotes cross-phosphorylation of key cytoplasmic tyrosine residues within RET’s intracellular domains. These domains subsequently recruit specific adaptor proteins which activate downstream signaling cascades involved in cellular proliferation and survival, such as the MAPK, PI3K, JAK-STAT, and PKA pathways (Figure 1) [14,15,16,17]. GFL and GFR family members have distinct tissue-specific expression patterns which play a crucial role in determining their biological functions.

## 3. Oncogenic *RET* Alterations

Oncogenic activation of RET can occur via two mechanisms: (1) germline or sporadic mutations activating RET’s kinase domain or (2) chromosomal rearrangements between *RET* and a gene partner typically containing a dimerization domain [18,19]. Both alterations can trigger oncogenic RET ligand-independent phosphorylation leading to constitutive activation of downstream transduction cascades which then results in morphological transformation and tumor growth. RET’s transforming potential was first described in 1985 using NIH/3T3 mouse fibroblastic cell lines transfected with rearranged lymphoma DNA containing a *RET* coding sequence [15]. Later, *RET* was one of the first oncogenes identified in solid tumors, more specifically in PTCs with *RET* fusions [20,21].

### 3.1. Activating RET Mutations: MTC

Germline or somatic activating *RET* mutations are enriched in patients with MTC. Pathognomonic non-synonymous point mutations in *RET*’s kinase or extracellular domain occur in this population. Mutations in the extracellular domain, such as those involving codon C634 [22], can lead to ligand-independent dimerization. In contrast, mutations in the kinase domain, such as RET M918T and the V804 gatekeeper mutations, can trigger monomeric activation of RET [23,24]. Inherited *RET* activating mutations are associated with MTC, either alone (familial medullary thyroid cancer [FMTC]) or as part of the multiple endocrine neoplasia type 2A (MEN 2A) or MEN 2B syndromes [23]. All MEN 2 syndromes are associated with an increased risk of MTC, pheochromocytoma, and neuroendocrine malignancies (including pulmonary or extra-pulmonary carcinoid tumors). Whereas patients with MEN 2A face an increased incidence of parathyroid hyperplasia and adenoma, patients with MEN 2B can develop intestinal ganglioneuromas or mucosal neuromas. FMTC, MEN 2A, and MEN 2B are all associated with a high penetrance of MTC in gene carriers and a 70–100% risk of developing MTC by age 70 [23]. Of note, approximately 2% of MEN 2A and 50% of MEN 2B cases arise from de novo germline mutations [25]. In contrast, somatic activating *RET* mutations are seen in 40–70% of sporadic MTCs, correlate with aggressive phenotype, and are most commonly associated with RET M918T mutations [26,27,28].

### 3.2. RET Rearrangements: Thyroid Cancer

As opposed to *RET* activating mutations, *RET* rearrangements are only known to be acquired somatically. Chromosomal rearrangements involve the intact 3′ tyrosine kinase domain of *RET* with various heterologous gene loci which provide dimerization domains for the constitutive expression of RET [29,30]. Fusion partners contribute different types of motifs to the fusion protein, including coiled-coil, LIS1 homology (LisH), tryptophan–aspartate repeat (WDR), and sterile alpha motif (SAM) domains, all of which mediate dimerization and ligand-independent signaling. Most *RET* rearrangements are intrachromosomal events, either paracentric or pericentric, involving upstream partners on chromosome 10 [29,31]. These include *CCDC6* or *NCOA4* [31], the most common gene partners in *RET*-rearranged PTC, as well as *KIF5B*, which is commonly involved in NSCLC [29]. Additional intrachromosomal gene partners include *ACBD5*, *ANKRD6*, *FRMD4A,* or *KIAA1217* [32,33,34,35,36]. Interchromosomal fusions or translocations are less frequent, but have been described with many partners, including *RELCH*, *ERC1,* and *TRIM33* [32,37,38].

*RET* rearrangements occur as a consequence of stochastic DNA double-strand breaks (DSBs) and non-homologous end joining. DSBs can result from environmental factors, such as ionizing radiation exposure, or endogenous mechanisms, including DNA replication stress or topological stress from topoisomerase activity [39,40,41]. Breakpoints typically occur at selected regions of the *RET* gene, often involving intron 11, which gives rise to a fusion protein containing only the cytosolic domain of RET [42]. Non-random recurrence of *RET* fusion partners is predisposed by the contiguity of specific loci from coiled chromosomes in the nucleus of normal cells. Spatial proximity of fusion partners involving linearly distant genes of the same chromosome or juxtaposed regions from two different chromosomes facilitates simultaneous double-end breaks and end joining [43,44].

In unselected PTC populations, *RET* fusions can be identified in 10–30% of patients [31,45,46,47,48]. *RET* fusions are typically mutually exclusive with additional oncogenic rearrangements such as *BRAF* alterations [49]. *RET* rearrangements represent a hallmark of ionizing radiation exposure, whether it be therapeutic or accidental, and were initially described in survivors of the Chernobyl nuclear disaster [50,51]. In such populations, the prevalence of *RET* rearrangements approximates 40–70% and is often seen in pediatric patients, likely as a consequence of increased follicular cell proliferation in this age group [50,51,52]. Of note, additional thyroid cancer subtypes can also be associated with *RET* fusions although at a lower prevalence, including poorly differentiated thyroid carcinoma (5%) and anaplastic thyroid cancer (1%) [48].

### 3.3. RET Rearrangements: NSCLC

The first case of *RET* rearrangement in a lung cancer patient was reported in 2012 by Ju et al. in a 33-year-old never-smoker with lung adenocarcinoma without known driver alterations in *EGFR*, *KRAS,* and *ALK*. The authors reported a novel fusion between *RET* and *KIF5B* which was identified in two additional patients in a replication study, thus defining a new molecular subset in patients with NSCLC [53]. The oncogenic potential of *KIF5B-RET* was later demonstrated in pre-clinical models via transfection of NIH/3T3 fibroblast cell lines with *KIF5B-RET*, leading to anchorage-independent growth activity [29].

In NSCLC, *RET* rearrangements occur at a frequency of approximately 1–2% in unselected patient populations and are generally non-overlapping with alterations in other oncogenic drivers, such as *EGFR*, *ALK,* and *ROS1* [46,47,48,54,55]. In this population, the most common fusion partner of *RET* is *KIF5B* (40–70%), followed by *CCDC6* (15–30%) [38,56,57,58]. The *KIF5B* gene partner is highly specific to NSCLC [48,59]. Patients diagnosed with *RET*-rearranged NSCLC typically have no history of tobacco smoking and tend to be relatively younger with a median age at diagnosis of <65 years [12,48,60,61]. When analyzing clinicopathologic features, *RET* fusions are mostly identified in adenocarcinomas, many of the solid or signet-ring cell subtypes, and typically present as peripheral tumors on lung imaging [54,55,62,63,64]. Brain metastases are common in *RET*-rearranged metastatic NSCLC, with a prevalence of 25% at diagnosis and a cumulative incidence of 45% during longitudinal follow-up [65]. Of note, *RET* fusions have also been described as an acquired resistance mechanism in patients with *EGFR-*mutant NSCLC treated with osimertinib [66]. In this setting, *RET* fusions typically involve gene partners other than *KIF5B* [59].

### 3.4. RET Activating Mutations across Other Histologies

Whereas activating *RET* mutations predominantly involve MTC, such alterationsthey have also been detected in other cancer populations. In a cohort of 4871 patients screened using targeted next-generation sequencing (NGS), Kato et al. described 88 patients (1.8%) presenting a *RET* aberration, with *RET* mutations (34/88) reported more commonly than *RET* fusions (27/88) or amplifications (22/88). The most prevalent activating mutation was RET M918T (*n* = 7), which apart from MTC was also reported across a spectrum of additional neuroendocrine tumors, including paraganglioma, pheochromocytoma, and carcinoid tumors [47]. Of note, small-cell lung cancer (SCLC), a high-grade neuroendocrine tumor, has also been associated with *RET* activating mutations although to a far lesser degree than nearly universal *TP53* and *RB* loss-of-function mutations [59,67].

In their study, Kato et al. identified other canonical activating *RET* mutations, including V804M (*n* = 4) and C634R (*n* = 2), which involved cases of breast carcinoma (C634R) or gastrointestinal malignancies (V804M) such as colorectal adenocarcinoma, gastrointestinal stromal tumor, or hepatocellular carcinoma [47]. Recurring non-canonical activating mutations included RET E511K (*n* = 3) which was identified in patients with anaplastic thyroid cancer, endometrial adenocarcinoma, and Merkel cell carcinoma. Additional mutations of unknown significance involving tumors other than MTC included RET R114H (*n* = 6), M1109I/T (*n* = 2), RET R525Q (*n* = 2), RET R600Q (*n* = 2), RET V706M (*n* = 2), RET A756V (*n* = 1), RET M255I (*n* = 1), RET R163Q (*n* = 1), and RET T636M (*n* = 1).

### 3.5. RET Fusions across Other Histologies

Outside of NSCLC and PTC, *RET* fusions have been identified in other cancer types at variable frequencies (Figure 2). Pancreatic acinar cell carcinoma (PACC), a rare subtype representing less than 2% of pancreatic malignancies [68], typically lacks mutations in *KRAS*, *CDKN2A,* and *TP53*, while being associated with recurrent gene rearrangements [69]. *BRAF* fusions represent a common oncogenic event in PACC and are present in 10–20% of cases [69]. In addition, *RET* rearrangements have been identified in 7.5% of patients with PACC using fluorescence in-situ hybridization (FISH), with *RET* and *BRAF* fusions being mutually exclusive [70].

*RET* fusions have also been found in patients with metastatic colorectal cancer [48,49,59,71], with *NCOA4* and *CCDC6* as the most common fusion partners. When compared to a control group of *RET* wild-type tumors, *RET* rearrangements were enriched in patients with *RAS* and *BRAF* wild-type right-sided primary tumors. *RET* fusions were also more common in microsatellite instability-high (MSI-high) tumors when compared to MSI-low tumors (odds ratio 12.1), while being mutually exclusive with additional genomic aberrations such as *RAS* and *BRAF*. In this study, *RET* fusions were independently associated with worse overall survival (OS) when compared to patients without *RET* fusions (OS; 14.0 vs. 38.0 months, *p* < 0.001, respectively) [71]. This series should be interpreted with caution as study findings predate immune checkpoint inhibitor (ICI) US FDA approval for patients with MSI-high tumors; ICI use in this subgroup is active and may impact survival outcomes.

Alterations involving the *RET* oncogene have also been observed in patients with breast cancer, including canonical fusions and non-canonical missense mutations or amplifications [48,49,59,72]. Preclinical models demonstrated the transforming capacity of both *RET* fusions and amplifications in breast cancer cell lines through activation of the MAPK and PI3K pathways. While most *RET* fusions involved HER2-negative breast cancers with variable estrogen receptor (ER) expression, presence of the *NCOA4*-*RET* fusion was also identified as a possible bypass pathway resistance mechanism to dual HER2-targeted therapy and aromatase inhibitor in a case of ER-positive, HER2-overexpressed breast cancer.

Salivary gland carcinoma subtypes, such as salivary intraductal carcinomas, have been associated with an enrichment for *RET* fusions [73]. *RET* rearrangements have also been documented in salivary gland mammary analogue secretory carcinomas (MASCs), which are characterized by recurrent *ETV6* rearrangements, with a minority of cases involving non-*NTRK* fusions, such as *RET* [74]. Apart from *ETV6-RET* fusions, *RET* rearrangements involving additional gene partners have also been described in salivary MASC such as the *VIM-RET* fusion [75].

Spitzoid neoplasms, including spitzoid melanomas and atypical spitzoid tumors, are associated with *RET* fusions in approximately 3% of cases. *RET* fusions typically involve the *KIF5B* gene partner and are mutually exclusive with other fusions such as *ROS1*, *NTRK1*, *ALK*, and *BRAF*. Interestingly, *RET* fusions were also detected in 3% of Spitz nevi, which represent uncommon, benign, melanocytic lesions typically arising in children and young adults. Such finding suggests that *RET* fusions may occur early in the pathogenesis of spitzoid malignancies [76].

*RET* rearrangements have been identified in sporadic cases of pheochromocytoma, without MEN 2 syndrome association. Interestingly, those include *RET* fusions with 3′ gene partners such as *RET-SEPTIN9* [77] and *RET-GRB2* [78]. In such cases, fusion events are associated with ligand-independent recruitment of canonical RET adaptor proteins, leading to constitutive activation of downstream signaling [78].

Pediatric spindle-cell mesenchymal neoplasms represent an additional group of tumors with recurrent oncogenic fusions, including *NTRK*, *BRAF*, *MET,* and *RET*. Such malignancies present histological overlap with so-called infantile fibrosarcoma cases and classically arise in the first 2 years of life. *RET*-rearranged spindle cell neoplasms were described with the *MYH10*, *KIAA1217,* and *CLIP2* gene partners and had clinical presentations ranging from localized resectable tumors to more aggressive metastatic disease [79].

Histiocytosis is a clonal disorder of hematopoietic cells with a wide spectrum of presentations, including Langerhans cell histiocytosis (LCH) and Erdheim–Chester disease. Histiocytic neoplasms are commonly driven by activating mutations in the MAPK pathway, including *BRAF* and *MEK* alterations. Cutaneous xanthogranuloma, a non-LCH histiocytosis, has been associated with the *NCOA4–RET* rearrangement [80].

Further studies using comprehensive sequencing methods have also revealed the presence of *RET* rearrangements in *KRAS* wild-type pancreatic ductal adenocarcinoma, stomach adenocarcinoma, esophageal cancer, cholangiocarcinoma, bladder carcinoma, head and neck cancer, mesothelioma, low-grade glioma, atypical lung carcinoid tumor, and chronic myeloproliferative neoplasms [48,59,81,82,83,84].

### 3.6. RET Amplifications

Additional cytogenetic events such as *RET* amplifications have been described in MTC, PTC, and anaplastic thyroid cancer [85,86], although their role in thyroid carcinogenesis remains questionable. In NSCLC, *RET* copy number gains (8.1%) and amplifications (2.8%; when defined as innumerable *RET* clusters, or ≥7 copies in >10% of tumor cells) are detected in a higher proportion of tumor samples when compared to *RET* rearrangements (0.7%) [87]. In a pan-cancer cohort, *RET* amplifications (defined as ≥6 copies of wild-type *RET*) were detected in 0.16% (145/91,466) of tumor samples, including 0.13% (15/11,622) of NSCLC cases [88]. *RET* amplifications have been observed in several other tumor types, including hepatobiliary cancers [88,89], prostate cancer [88], breast cancer [77], glioblastoma [90], stomach adenocarcinoma [91], colorectal adenocarcinoma [91], and bladder urothelial carcinoma [91]. The clinical significance of *RET* amplifications and their correlation with increased RET protein expression have not been fully characterized.

## 4. Standard Methods for Detection

Since RET-selective targeted therapies are US FDA approved or available on clinical trials in countries without approval, universal screening for *RET* alterations should be considered when available, including for patients with NSCLC, thyroid cancer, and those with tumor histologies known to be enriched with *RET* fusions [92]. Activating *RET* mutations can be easily detected with the use of NGS. Reverse-transcriptase polymerase chain reaction (RT-PCR) or Sanger sequencing can alternatively be used in clinical settings where NGS is not readily available. Oncogenic *RET* rearrangements can also be identified using alternative testing methods, including FISH [53]. RET diagnostic assays, however, demonstrate different performance characteristics, including variable sensitivity for capturing specific *RET* fusions and mutations [48]. Initial clinical studies predominantly used RT-PCR and FISH techniques to detect *RET* rearrangements in tumor samples. However, the emergence of more comprehensive sequencing platforms such as NGS offered the added potential of identifying upstream partner genes involved in *RET* rearrangements, as well as confirmation of pathogenicity of *RET* fusions, and assessment of concurrent genomic alterations [93].

### 4.1. Immunohistochemistry (IHC)

Given that IHC can be used to measure RET protein overexpression, studies have explored the diagnostic utility of RET IHC as a potential screening assay for oncogenic *RET* alterations, particularly *RET* fusions. In clinical practice, RET IHC is, however, seldom performed as immunostaining techniques and antibodies remain poorly standardized [48,94]. Using a cytoplasmic staining cutoff of ≥1% of tumor cells, overall sensitivity and specificity of IHC with the EPR2871 clone is 87% and 82%, respectively [48]. Sensitivity of RET IHC differs according to fusion partner, with a 100% sensitivity for *KIF5B*, but only 50% sensitivity for *RET* rearrangements involving the *NCOA4* fusion partner [48]. It should be noted that wild-type RET expression can be detected in normal cells by IHC, including in parafollicular C cells [94], tracheal epithelium [95], adrenal glands [96], and colorectal tissue [97]. A study also demonstrated that RET expression can be elevated in NSCLC even in the absence of *RET* fusions [98]. Consequently, RET IHC may be misleading when used as a standalone test, and alternative diagnostic assays usually outperform IHC as screening methods for *RET* alterations [74,94]. Given these challenges, RET IHC is not recommended as a clinical screening assay for oncogenic *RET* alterations [94].

As previously alluded to, solid tumors associated with an enrichment in *NTRK* fusions, such as MASC, infantile fibrosarcoma, and spitzoid neoplasms, can alternatively be associated with *RET* rearrangements upon negative pan-Trk IHC, which represents a validated screening tool for *NTRK* rearrangement [99]. Pan-TrK negativity in the setting of tumor histologies with a higher prevalence of kinase fusions should thus trigger *RET*-specific diagnostic assays due to therapeutic considerations.

### 4.2. FISH

FISH represents an alternative method for detection of *RET* rearrangements which was more commonly used before DNA-based NGS became readily accessible. Break-apart FISH is typically considered to be positive if ≥10–15% of tumor cells demonstrate a rearrangement pattern characterized by separation of 2 signals flanking the *RET* gene when evaluating 100 non-overlapping nuclei [48,94]. While the advantages of FISH include single-cell resolution as well as rapid turnaround time, there are important limitations to this method. First, FISH analysis does not have the spatial resolution to provide information regarding exonic breakpoints or inclusion of the RET kinase domain in the *RET* fusion. These datapoints are critical for confirming whether the *RET* rearrangement is pathogenic, and therefore actionable. Second, the accuracy of FISH depends on the involved fusion partner. Break-apart FISH can detect intrachromosomal rearrangements [71] with a 100% sensitivity for *KIF5B* and *CCDC6*, but is less sensitive for *NCOA4* rearrangements (67% sensitivity) [48]. Such differences are likely attributable to the close contiguity of *RET* and *NCOA4* on chromosome 10 which can lead to a more subtle splitting pattern [100]. When considering tumor primaries, sensitivity is highest with NSCLC (100%), followed by thyroid cancer (88%) and additional tumor types (75%), with such differences likely driven by the respective prevalence of fusions such as *NCOA4-RET* [48].

### 4.3. DNA PCR-Based Assays

Sanger sequencing, also known as chain-termination sequencing, allows robust and reliable identification of known single-nucleotide alterations when present at a relative allelic frequency above 15% [94]. However, due to its inability to detect gene rearrangements, partial gene deletions, and alterations with low variant allele frequencies (VAFs), it has limited use outside of FMTC and MEN2A diagnostics, where it still plays a role as a confirmatory test.

Quantitative PCR on DNA allows reliable detection of hot-spot mutations present at VAFs as low as 1% [94]. The finite number of primers used with this technique, however, plays a key role in limiting the reliability of these assays in detecting fusions and rarer oncogenic mutations that occur outside of hot spots [94]. While NGS-based assays are preferable for detecting *RET* activating mutations in clinical practice, quantitative PCR can still be considered as a viable alternative if NGS is not readily available.

### 4.4. NGS: DNA

Advancements in DNA sequencing techniques have facilitated the detection of *RET* alterations across various tumor types. Targeted DNA-based NGS can be used for simultaneous detection of *RET* mutations and rearrangements given sufficient exonic coverage and inclusion of introns (for *RET* fusions). A number of gene panels are clinically available to detect canonical *RET* alterations, with variable detection rates for *RET* alterations. When evaluating the performance of MSK IMPACT (hybrid capture DNA-based NGS panel) compared to RNA-based NGS as a reference standard (MSK-Fusion) in detecting *RET* fusions, MSK-IMPACT was highly sensitive; however, there were select cases of rearrangements of unknown significance (i.e., non-canonical *RET* rearrangements) where only RNA sequencing was able to confirm the presence or absence of oncogenic *RET* fusion transcripts [48]. Due to its high sensitivity, DNA-based NGS also allows for the identification of somatic mutations occurring at low VAFs. Subclonal *RET* rearrangements, which are only present in a minority of cells, can be detected by DNA-based NGS and are thought to reflect inherent tumor heterogeneity, including cases of acquired *RET* alterations as a resistance mechanism to osimertinib in *EGFR-*mutant NSCLC [31,101,102].

Plasma-based circulating free DNA (cfDNA) allows for non-invasive detection of *RET* alterations and represents an alternative to tissue-based DNA sequencing. The diagnostic yield of cfDNA testing is, however, contingent on the tumor burden and DNA shedding rates at the time of liquid biopsy, with increased sensitivity in patients with progressive or metastatic disease compared to those with stable or localized disease [48]. In addition to variable sensitivity, limited specificity associated with DNA-based sequencing (i.e., detection of *RET* rearrangements of unknown significance) remains an issue for cfDNA-based liquid biopsy testing. The use of cfDNA, however, can capture tumor heterogeneity and dynamics of treatment responses with longitudinal testing, thus providing a valuable platform to detect the influence of clonal and subclonal mutations on tumor progression [59]. Prospective cfDNA testing can therefore provide valuable insight when evaluating on-treatment resistance mechanisms and exploring rational combination therapies.

### 4.5. NGS: RNA

RNA-based NGS allows for the identification of functionally definitive *RET* fusion transcripts, irrespective of the underlying DNA-level events. In many clinical settings, RNA-based testing is an important confirmatory assay and often considered the gold standard method for *RET* fusion detection [103]. In general, RNA sequencing can be integrated into clinical workflow in two different ways. In workflows that involve screening by DNA-based sequencing, RNA testing can be performed on select cases with *RET* rearrangements of unknown significance to provide a more definitive and functional assessment. In addition, RNA-sequencing can be performed upfront with DNA-sequencing, although this may not be necessary in the majority of cases since DNA-sequencing is quite sensitive for *RET* fusion detection. As formalin fixation and block preservation of tissue is associated with significant RNA degradation, quality control of preanalytical conditions is crucial to ensure results accuracy.

## 5. *RET*-Altered Cancers: Pre-Targeted Therapy Era

The prognostic significance of *RET* fusions remains a matter of debate, with two Chinese studies delineating *RET* as a biomarker of inferior OS in NSCLC, especially in female patients [55,104]. Prior to the advent of selective *RET* inhibition, patients with *RET*-rearranged NSCLC treated with conventional therapies in the United States, including chemotherapy and immunotherapy, demonstrated similar survival outcomes when compared to patients with *RET* wild-type NSCLC in a retrospective analysis [60]. When considering chemotherapy options, pemetrexed alone or as part of a combination regimen can achieve durable clinical benefit in patients with *RET*-rearranged NSCLC. In a retrospective study, median progression-free survival (PFS) on pemetrexed-inclusive regimens was 19 months in *RET*-rearranged NSCLC, comparable to historical data from patients with tumors harboring *ALK* or *ROS1* fusions [105].

As patients with *RET*-rearranged NSCLC were not specifically excluded from registrational ICI trials, a unicentric retrospective analysis was conducted in order to characterize treatment responses in this population. A majority (81%) of *RET*-rearranged patients had either a PD-L1 score of 0% or low expression of PD-L1 (1–49%). In addition, tumor mutational burden (TMB) was significantly lower when compared to *RET* wild-type cancers (1.75 vs. 5.27 mutations/megabase, respectively), reflecting the presence of an oncogenic driver. Suboptimal outcomes were observed in this cohort, with no objective responses seen in 13 evaluable patients treated with anti-PD(L)-1 with or without anti-CTLA4 therapy. Of note, baseline immunophenotype did not correlate with responses to ICI activity, and the median PFS was 3.4 months, consistent with *RET* fusion-positive lung cancers being immunologically cold tumors [57].

Little data are available on the efficacy of chemoimmunotherapy in patients with *RET*-altered cancers. In patients with *RET* fusion-positive NSCLCs, the combination of platinum-based chemotherapy and pembrolizumab in the first-line setting was associated with inferior OS and PFS when compared to pralsetinib in a retrospective analysis. Prospective data will be generated by the first line AcceleRET and LIBRETTO-431 studies that randomize patients with *RET* fusion-positive lung cancers to selective RET inhibitor therapy or chemotherapy +/− immunotherapy.

## 6. Multikinase RET Inhibitors

The efficacy of multikinase inhibitors (MKIs), including cabozantinib, sorafenib, vandetanib, and lenvatinib, which all exhibit activity against RET, has been observed in vitro with variable half maximal inhibitory concentration (IC_50_) in kinase assays [61]. Pre-clinical modeling studying *RET* fusion driven cancer cell lines and patient-derived xenograft (PDX) models first confirmed the activity of MKIs through decreased RET autophosphorylation and inhibition of cell proliferation [35,106,107,108,109,110,111]. Similar biological activity was replicated in models with activating *RET* mutations, including MTC cell lines and PDX models, and in PDX models with *RET* amplifications [109,111,112,113], providing insights into the druggability of *RET* alterations by MKIs initially designed for other indications. Prospective trials have since provided evidence of confirmed responses and sustained disease control in a proportion of *RET*-altered thyroid, lung, and breast cancer patients treated with RET MKIs.

### 6.1. Cabozantinib

Cabozantinib is a potent MKI which exhibits significant inhibitory activity against various kinases, including RET (with an IC_50_ value of 5.2 nM), as well as ROS1, MET, VEGFR2, AXL, TIE2, and KIT [64]. Cabozantinib was approved by the US FDA for the treatment of progressive metastatic MTC in 2012, following publication of a landmark phase 3 trial in a biomarker-unselected population [114]. Correlative analyses in this population later demonstrated increased PFS benefit in patients with *RET* mutations when compared to wild-type *RET* [115]. In 2013, rapid translational research led to the first clinical validation of cabozantinib use in a biomarker-selected population with *RET*-altered tumors, less than 2 years following initial identification of *RET* fusions as oncogenic drivers in patients with NSCLC [30,32,56]. A phase 2 trial enrolling patients with NSCLC and recurrent gene fusions first demonstrated evidence of clinical benefit with cabozantinib in three patients with *RET* fusions, with partial responses in two patients and prolonged disease stability in one patient [32,116].

The activity of cabozantinib as a *RET*-directed therapy is less characterized in additional cancer sites, such as patients with *RET* fusion-positive PTC. In a patient with ER+/HER2+ breast cancer harboring a *NCOA4-RET* fusion, treatment with cabozantinib in combination with trastuzumab and exemestane led to radiological tumor regression and symptomatic improvement following initial progression on the pertuzumab, trastuzumab, and anastrozole combination [72]. Whereas it remains challenging to determine with certainty whether the response was solely attributed to cabozantinib or a combination of these therapies, addition of a RET inhibitor led to a clinical response in this patient with treatment-refractory ER+/HER2-overexpressing breast cancer presenting a *NCOA4-RET* fusion [72].

### 6.2. Vandetanib

Phase 2 trials of vandetanib in pretreated patients with advanced *RET*-rearranged NSCLC demonstrated variable anticancer activity with overall response rates (ORRs) ranging from 18% to 50% [117]. In a biomarker-unselected population of patients with metastatic MTC, a placebo-controlled phase 3 trial demonstrated a PFS benefit with vandetanib (21 vs. 8 months with placebo) [118,119]. Of note, the correlation between survival outcomes and *RET* mutational status was not clearly evaluable in post-hoc analysis due to the low number of patients with wild-type *RET* [120]. In the setting of NSCLC, a retrospective analysis of four randomized phase 3 trials failed to demonstrate differential benefit with vandetanib in patients with *RET* amplifications or copy number gains when compared to the *RET* wild-type population [87]. In addition, no clinical benefit was seen in patients with *RET* rearrangements, although such alterations were detected at a low prevalence in this population [87]. Reports studying the use of vandetanib in biomarker-selected populations otherwise remain limited outside of the NSCLC population. Although rare, acquired *RET* mutations mediating resistance to *RET* MKIs such as vandetanib have been documented in the clinical setting, including the RET V804M [121] and V804L [122] gatekeeper mutations [122,123].

### 6.3. Limitations of Multitargeted RET Inhibitors

Studies involving additional RET MKIs, such as sorafenib [124], lenvatinib [34], and alectinib [125], also led to objective responses in *RET*-altered NSCLC, although with limited activity and prohibitive toxicity profiles [126]. In comparison to targeted therapies tailored to other oncogenic drivers commonly found in solid tumors, such as sensitizing *EGFR* or *BRAF V600E* mutations, as well as *ALK* or *ROS1* rearrangements, response rates to RET-directed therapy remained relatively modest until the recent advent of selective RET inhibitors.

Factors accounting for the discrepancy between the in vitro and clinical activity of RET MKIs include inherent pharmacokinetic properties as well as off-target toxicities limiting optimal dose intensity of RET inhibitors [61]. RET MKIs do not consistently achieve concentrations allowing for effective RET inhibition, with high IC_50_ values for both vandetanib and cabozantinib [122]. When considering metastatic disease to the brain, outcomes on RET MKIs were suboptimal, with an intracranial response rate of 18% in a cohort of patients treated with either cabozantinib, vandetanib, ponatinib, or alectinib [65], also highlighting the need for novel RET inhibitors with improved brain penetrance. RET MKIs can also be associated with a prohibitive safety profile, predominantly through VEGFR-mediated antiangiogenic toxicities [22]. Many of the MKIs targeting RET, including cabozantinib and vandetanib, are associated with preferential inhibition of VEGF signaling, with cabozantinib possessing a 100-fold higher anti-VEGFR potency relative to its anti-RET activity [112,127,128]. Treatment-emergent side effects from off-target VEGF inhibition, including hypertension, palmar-plantar dysesthesia, and proteinuria, can limit chronic drug dosing while impacting patients’ quality of life.

With the aim to improve RET selectivity, the VEGFR-sparing RET MKI RXDX-105 was studied in a phase 1/1b trial, but only demonstrated activity against non-*KIF5B*-associated tumors, thus limiting further clinical development [129]. Furthermore, RXDX-105 still demonstrated off-target toxicities through inhibition of non-RET kinases, with rash, fatigue, diarrhea, and hyperbilirubinemia reported as dose-limiting toxicities [129]. Unique safety signals also included drug hypersensitivity reactions, including three cases of drug reaction with eosinophilia and systemic symptoms (DRESS), likely attributable to RXDX-105 [129].

## 7. Selective RET Inhibitors

Mechanistically, RET inhibitors can be classified according to the Roskoski classification [130] which considers the unique chemical characteristics of each inhibitor, as well as the size of the molecule and the tridimensional conformation of the target enzyme in complex with the inhibitor. Kinases such as RET contain highly conserved structures, such as an activation loop (A-loop), an alphaC helix, and the Asp-Phe-Gly (DFG) motif [131], which play central roles in regulating kinase activation and substrate phosphorylation [132,133]. Phosphorylation of tyrosine residues in the A-loop activates the kinase and induces the opening (A-loop-out configuration) of the substrate binding pocket, while in the inactive state this structure is in the closed (A-loop-in) configuration [134]. The DFG motif is essential for ATP binding and effective catalysis and can adopt either the inactive (DFG-Dout) or the active (DFG-Din) conformation, with this latter state allowing phosphate transfer to the substrate [132,133]. Similarly, kinase activation induces conformational changes in alphaC helix which changes to the alphaC-in active configuration from the alphaC-out inactive configuration. These conformational changes induce dramatic modifications in protein structure accessibility for inhibitor binding [130]. An active RET kinase is characterized by the DFG-Din conformation in conjunction with the alphaC-in and A-loop-out configuration, while all other configurations in these regulatory structures constitute inactive states [130].

Aside from the conformational changes described above, several amino acidic residues can also contribute to volume accessibility for inhibitor binding [135], two of the most notable being the gatekeeper and the solvent front residues [108]. Bulky gatekeeper residue mutations can block access to RET kinase back pocket and modify ATP affinity [108], while solvent front mutations can restrict access to the solvent front cleft [136]. All clinically relevant RET inhibitors are ATP-competitive and can be classified as either type I or type II kinase inhibitors based on their kinase binding mode. Type I inhibitors such as vandetanib [137], pralsetinib [138], or selpercatinib [138] display affinity for the ATP binding pocket of active state RET and exert their inhibitory action via competition with ATP [139]. Type II inhibitors, such as cabozantinib, display affinity for RET in its inactive DFG-Dout conformation. Notably, the DFG-Dout conformation displays a bigger and more accessible back pocket compared to the active kinase, which is exploited for drug binding [118,139,140]. As previously alluded to, cabozantinib and vandetanib, however, both target a spectrum of kinases apart from RET, including VEGFR, EGFR, and KIT, often at a higher potency. Novel, selective RET inhibitors, including selpercatinib and pralsetinib, were designed to overcome such limitations while retaining activity against most *RET* alterations, including predicted gatekeeper resistance mutations [141].

### 7.1. Selpercatinib (LOXO-292)

Selpercatinib (LOXO-292) is an ATP-competitive, potent, and selective RET inhibitor with favorable pharmacokinetic properties and central nervous system (CNS) coverage. In contrast to RET MKIs, selpercatinib was specifically designed to target diverse *RET* fusions and *RET* activating mutations at nanomolar potency, including predicted acquired MKI resistance mutations. In a study using engineered cell lines and PDX models harboring *KIF5B-RET*, RET M918T, or RET V804M alterations, selpercatinib induced significant tumor regressions across models. Two patients with *RET*-altered malignancies refractory to MKI therapy received selpercatinib on single-patient protocols: a patient with RET M918T-mutant medullary thyroid cancer presenting an acquired RET V804M gatekeeper mutation and a second patient with a *KIF5B-RET* fusion-positive NSCLC and symptomatic brain metastases. Both patients had confirmed tumor responses, including a dramatic decrease in intracranial disease for the second patient, thus providing proof-of-concept of the activity of selective RET inhibition [121].

LIBRETTO-001 (NCT03157128) was a registrational phase 1/2 open-label basket study of selpercatinib in patients with solid tumors harboring oncogenic *RET* alterations [37,38,84]. The recommended phase 2 dose (RP2D) of selpercatinib was identified as 160 mg orally twice daily. In the NSCLC cohort, patients with prior disease progression on platinum-based chemotherapy demonstrated an ORR of 61%, whereas in previously untreated patients, the ORR was 84%, with rapid and durable responses seen in both subgroups (median duration of response 20–29 months) [142]. Of note, responses were seen regardless of the underlying *RET* fusion partner [38]. In patients with brain metastases, selpercatinib had an intracranial ORR of 82% and no CNS progression events were observed in patients without baseline intracranial involvement [143,144]. The thyroid cancer cohort included both *RET*-mutant MTC and *RET*-rearranged thyroid cancer. *RET*-mutant MTC patients previously treated with cabozantinib and/or vandetanib had an ORR of 69%, while responses were seen in 73% of patients without prior MKI therapy [37]. In previously treated *RET* fusion-positive thyroid cancer patients, most of which of papillary or poorly differentiated histology, the ORR was 64%. Converging real-world data in the MTC population with comparable results have subsequently been published [145]. Results from LIBRETTO-001 led to the US FDA regulatory approval of selpercatinib for the treatment of patients with *RET* fusion-positive NSCLC, *RET* fusion-positive thyroid cancers, and *RET*-mutant MTC in 2022 [146].

Importantly, the trial generated data in *RET*-rearranged, non-NSCLC, and non-thyroid cancer patients (Table 1). The ORR with selpercatinib was 44% in 45 patients with *RET* fusion-positive advanced solid tumors who previously progressed on standard of care therapies [84]. In this tumor-agnostic population, objective responses were documented in patients with refractory pancreatic, salivary gland, carcinoid, breast, small intestine, colorectal, ovarian, bile duct, and mesenchymal tumors, often with significant durability (median duration of treatment of 11 months). Most patients (58%) in the study had gastrointestinal malignancies and *NCOA4* was the most common fusion partner, identified in 38% of cases. When considering tumor heterogeneity, response rates were highest in the 11 patients with pancreatic cancer (ORR 55%) and the 4 patients with salivary gland primaries (ORR 50%). In contrast, the ORR was lowest in colorectal cancer patients (ORR 20%), possibly as a consequence of underlying genetic diversity and MSI [71].

Since the publication of the tumor-agnostic cohort of LIBRETTO-001, additional case reports of activity in non-NSCLC and non-thyroid *RET*-altered cancers have emerged, including in tumor types which were not previously represented in the LIBRETTO-001 data set. Selpercatinib led to rapid responses in a patient with non-hereditary malignant pheochromocytoma harboring a novel *RET-SEPTIN9* fusion [77]. Effective treatment with selpercatinib was also documented in *RET* fusion-positive pulmonary large cell neuroendocrine carcinoma [147] and ER+/HER2- breast cancer [148]. In addition, cutaneous xanthogranuloma, a non-LCH histiocytosis, has been associated with the *NCOA4–RET* rearrangement and treatment with selpercatinib led to a dramatic clinical response of cutaneous lesions [80].

Patient inclusion in the registrational studies of selective RET inhibitors was limited to solid tumors with *RET* fusions for the NSCLC and tumor agnostic cohorts. Subsequent case reports have shed light on the efficacy of selpercatinib in patients with *RET* amplifications. Notably, a patient with *RET*-amplified (≥6 copies) NSCLC exhibited systemic and intracranial radiologic responses on selpercatinib following disease progression on chemoradiation and durvalumab [88]. In addition, a documented response to selpercatinib was observed in a patient with *RET*-amplified glioblastoma. Follow-up MRI at 3 months revealed near complete resolution of gadolinium-enhancing brain lesions, with a sustained response spanning over 8 months [90]. These reports underscore the potential of selpercatinib as a therapeutic option for patients with *RET*-amplified tumors, thus warranting further investigation and clinical exploration. 

Across cohorts, selpercatinib’s safety profile was found to be manageable. Commonly reported grade ≥ 3 toxicities included the following: hypertension (20–22%), increased alanine aminotransferase (ALT; 11–16%), increased aspartate aminotransferase (AST; 9–13%), hyponatremia (6–8%), diarrhea (6%), and QT interval prolongation on electrocardiogram (5%) [37,84,142]. Most of these treatment-related adverse events (AEs) did not warrant dose interruptions or reductions. In fact, only 2% of patients discontinued treatment due to drug-related toxicities in LIBRETTO-001, consistent with selpercatinib’s improved selectivity profile [37,38].

Other treatment-emergent toxicities have, however, been identified in the setting of selpercatinib therapy. Prior use of ICI was associated with an increased risk of hypersensitivity reaction to selpercatinib [149]. In addition, cases of spontaneous chylous effusions have been documented retrospectively, occurring in 7% of patients treated with selpercatinib. Both chylothorax and chylous ascites have been identified, with 76–80% of patients requiring additional therapeutic drainage after initial diagnostic thoraco- or paracentesis. Prospective recognition of such a side effect is crucial in clinical practice in order to avoid misattribution of chylous effusions to cancer progressive events [150]. Selpercatinib has also been associated with treatment-emergent hypothyroidism through off-target inhibition of type 2 iodothyronine deiodinase, which mediates active triiodothyronine (T3) synthesis from the inactive thyroxine (T4) form. Substituting levothyroxine (synthetic T4) with liothyronine (synthetic T3) supplementation in athyreotic patients normalized T3 levels and restored euthyroidism in this population [151].

### 7.2. Pralsetinib (BLU-667)

Pralsetinib (BLU-667) is a second, potent, ATP-competitive, selective RET TKI which has been approved by the FDA for the treatment of *RET* fusion-positive NSCLC and *RET*-altered thyroid cancers based on the results of the registrational ARROW trial [152,153]. In this multicohort, open-label, phase 1/2 trial, patients with locally advanced or metastatic solid tumors harboring a *RET* alteration were treated with pralsetinib as part of a dose-escalation design, with 400 mg once daily identified as the RP2D. In the NSCLC cohort, patients with prior platinum-based chemotherapy had an ORR of 59% and the ORR was 72% in previously untreated patients. In patients with measurable brain metastases at baseline, the intracranial response rate was 70% [58]. Real-world data later confirmed the superiority of pralsetinib over first-line ICI plus or minus chemotherapy in *RET* fusion-positive NSCLC [154]. The thyroid cancer cohort of ARROW included patients with *RET*-mutant MTC, either previously untreated or with progression on cabozantinib and/or vandetanib, as well as patients with previously treated *RET*-rearranged thyroid cancer. The following ORRs were observed: 71% in treatment-naïve MTC patients, 60% in MTC patients with prior cabozantinib and/or vandetanib, and 89% in previously treated *RET* fusion-positive thyroid cancer patients [153].

In patients with non-NSCLC, non-thyroid cancers, the tumor agnostic cohort of the ARROW trial demonstrated an ORR of 57% across histologies (Table 1). Pan-cancer efficacy was seen regardless of solid tumor types, including in epidermal (pancreatic, bile duct, head and neck, gastric) and mesenchymal (soft tissue sarcoma) tumors [36]. Of note, both complete and partial responses were observed in four pancreatic carcinoma cases (17% of the cohort). Such findings are in line with additional case reports of dramatic responses to pralsetinib in *RET*-rearranged pancreatic cancer, a tumor site associated with intrinsic refractoriness to most therapeutics [155]. Reponses to pralsetinib in *RET*-rearranged tumors have also been documented in patients with sarcomatoid carcinoma [156], as well as triple-negative breast cancer [157].

**Table 1 cancers-15-04146-t001:** Efficacy of selective RET inhibitors in non-NSCLC, non-thyroid cohorts with *RET* fusions.

Selpercatinib						
Study	Study design	Tumor site	Patients, *n*	ORR, *n* (%)	Responses (*n*)	DOR range (mo)
LIBRETTO-001 trial (2022) [84]	Prospective	Pancreas	12	6 (55%)	PR (6), SD (6)	3–38 *
		Colorectal	10	2 (20%)	PR (2), SD (8)	6–13
		Salivary	4	2 (50%)	PR (2), SD (2)	6–29 *
		Unknown primary	3	1 (33%)	PR (1), SD (2)	9 *
		Breast	2	2 (100%)	CR (1), PR (1)	2–17
		Sarcoma	2	2 (100%)	PR (2)	15 *
		Xanthogranuloma	2	NE	N/A	N/A
		Carcinoid (lung)	1	1 (100%)	PR (1)	24 *
		Skin carcinoma	1	NE	N/A	N/A
		Cholangiocarcinoma	1	1 (100%)	PR (1)	6 *
		Ovarian	1	1 (100%)	PR (1)	15 *
		Pulmonary carcinosarcoma	1	NE	N/A	N/A
		Neuroendocrine (rectum)	1	NE	N/A	N/A
		Small intestine	1	1 (100%)	CR (1)	25
Durham et al. (2019) [80]	Case report	Xanthogranuloma	1	N/A	Clinical response	N/A
Kander et al. (2021) [81]	Case report	Carcinoid (lung)	1	N/A	PR	6 *
Watanabe et al. (2021) [148]	Case report	Breast, ER+/HER2-	1	N/A	CR	10 *
Mweempwa et al. (2021) [77]	Case report	Pheochromocytoma	1	N/A	PR	5 *
Arora et al. (2023) [147]	Case report	Neuroendocrine (lung)	1	N/A	PR	12 *
**Pralsetinib**						
ARROW trial (2022) [36,58]	Prospective	Pancreas	4	4 (100%)	CR (1), PR (3)	3–27 *
		Cholangiocarcinoma	3	2 (67%)	PR (2), SD (1)	8–19
		Neuroendocrine	3	2 (67%)	PR (2), SD (1)	11–14 *
		Sarcoma	3	2 (67%)	CR (1), PR (1), SD (1)	11
		Head and neck	2	1 (50%	PR (1), SD (1)	9
		Colorectal	2	0 (0%)	SD (2)	N/A
		Small cell lung cancer	2	1 (50%)	PR (1), SD (1)	9 *
		Unknown primary	1	1 (100%)	CR (1)	5
		Stomach	1	0 (0%)	PD (1)	N/A
		Ovarian	1	0 (0%	PD (1)	N/A
		Thymic	1	0 (0%)	SD (1)	N/A
Wu et al. (2022) [156]	Case report	Sarcomatoid carcinoma (lung)	1	N/A	PR	NE
Zhang et al. (2023) [155]	Case report	Pancreas	1	N/A	PR	12 *
Zhao et al. (2023) [157]	Case report	Breast, triple negative	1	N/A	PR	8 *

ORR: objective response rate; DOR: duration of response; Mo: months; CR: complete response; PR: partial response; PD: progressive disease; NE: non-evaluable; N/A: not applicable. * Ongoing response.

Across studies, pralsetinib was well tolerated with common grade 3–4 AEs consisting of neutropenia (13–31%), hypertension (7–17%), anemia (10–14%), lymphopenia (9–12%), and increased creatine phosphokinase (CPK; 6%) [36,58,153]. Additional serious AEs, including treatment-emergent pneumonitis (4%), were recorded on pralsetinib [153].

## 8. Acquired Resistance

Limited accounts of *RET* acquired resistance mutations were reported during the era of RET MKIs [121,122,129,158], likely indicative of suboptimal target inhibition and restricted selective pressure on *RET*-altered clones. Since the US FDA approval of potent and selective RET inhibitors, emergence of concurrent genetic alterations has been documented in patients with baseline *RET* aberrations with disease progression on selpercatinib or pralsetinib. On-target *RET* solvent front mutations involving the G810 residue (G810 C/S/R) represent a recurrent mechanism of resistance, leading to steric hindrance to the binding of selpercatinib [136,159]. RET G810X mutations also mediate cross-resistance to pralsetinib [138,160]. Alternative roof solvent front mutations, such as RET L730V/I, have been identified in patients with pralsetinib-resistant tumors, while retaining sensitivity to selpercatinib in preclinical models [161]. On-target mutations affecting the Y806 or V738 residues have also been identified as mediators of resistance to both selpercatinib and pralsetinib [138]. 

Off-target resistance mechanisms have been observed in patients treated with selpercatinib or pralsetinib, including bypass pathway activation involving *MET* amplifications [159]. Proof of concept clinical reports have demonstrated the activity of MET inhibitors in the setting of acquired *MET* amplification in patients with *RET* fusion-positive NSCLCs. The combination of crizotinib and selpercatinib led to responses in selpercatinib-refractory patients with acquired *MET* amplification. In a case report, sequential use of crizotinib and capmatinib led to a response in a patient with *KIF5B-RET* fusion-positive NSCLC with acquired *MET* amplification who had initially progressed on pralsetinib. Downstream MAPK pathway reactivation has also been involved in patients with acquired resistance to selective RET inhibitors [140,159,162], including the emergence of activating *KRAS*, *NRAS,* or *BRAF* mutations [162]. In addition, histological transformation to SCLC has been described as an acquired resistance mechanism to pralsetinib in patients with *RET-*rearranged lung adenocarcinoma [163,164].

## 9. Conclusions and Future Directions

In conclusion, *RET* fusions and *RET* activating mutations are therapeutically actionable oncogenic drivers which can be effectively targeted by novel and potent inhibitors selectively targeting the RET kinase. Deep and durable responses have been demonstrated both with selpercatinib and pralsetinib across a wide range of histologies, arguing for the universal screening of solid tumors for the presence of *RET* alterations. Alternatively, a two-step approach would be reasonable, with screening for *RET* alterations limited to patients without oncogenic drivers on initial testing. Next-generation RET inhibitors that potentially circumvent mechanisms of acquired resistance to selective RET inhibitors, including emergent solvent front mutations, are being explored. Ongoing clinical trials are assessing the safety and efficacy of novel RET inhibitors, including vepafestinib (NCT04683250) and LOXO-260 (NCT05241834). Additional efforts should lie in exploring the therapeutic potential of these newer-generation TKIs across different cancer types and defining the spectrum of *RET* alterations.

## Figures and Tables

**Figure 1 cancers-15-04146-f001:**
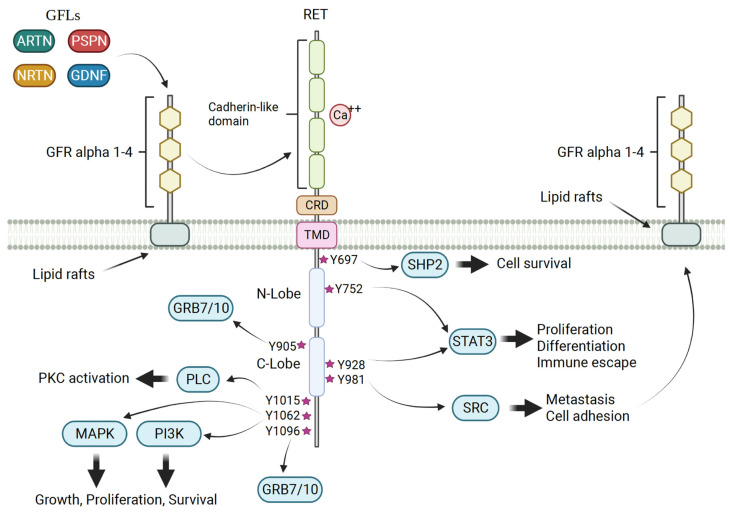
Canonical RET signaling. Activation of RET occurs when glial cell line-derived neurotrophic factor (GDNF) family ligands (GFLs) such as artemin (ARTN), neurturin (NRTN), persephin (PSPN), and GDNF bind to the co-receptor GDNF family receptor (GFR) alpha 1–4. Concurrently, calcium ions bind to the cadherin-like calcium binding domain, triggering the recruitment of RET and formation of the RET–GFR alpha complex. This complex brings together two RET monomers (not shown), leading to homodimerization and subsequent cross-phosphorylation of crucial tyrosine residues within RET (red stars). These phosphorylated tyrosine residues serve as docking sites for adaptor proteins involved in the propagation of RET signaling, including MAPK, PI3K/AKT, PLCγ, and SRC (which requires association with lipid rafts for activation). Consequently, activation of RET signaling promotes cell proliferation, growth, and survival by engaging multiple downstream signaling cascades.

**Figure 2 cancers-15-04146-f002:**
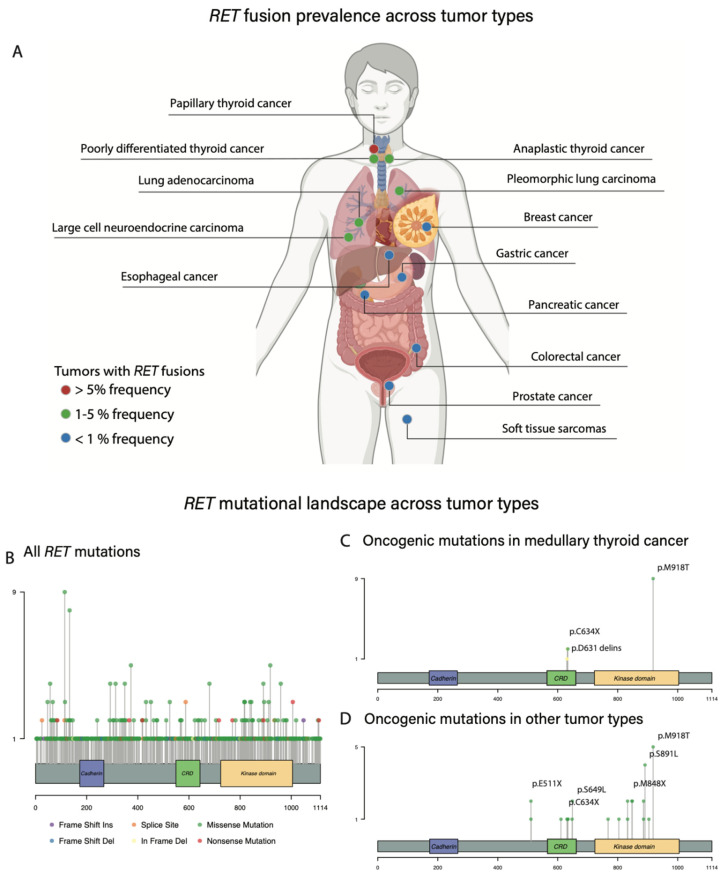
*RET* alterations across tumor sites. (**A**): *RET* fusions are observed in a variety of pediatric and adult cancer histologies. The prevalence of these fusions ranges from less than 1% in cancers of the breast, oesophagus, stomach, pancreas, colon/rectum, prostate, and soft tissue (represented in blue circles) to 1–5% in cancers of the thyroid gland (anaplastic or poorly differentiated) and the lung (pleomorphic, adenocarcinoma, large cell neuroendocrine) (represented in green circles). In addition, *RET* fusions are present in >5% of papillary thyroid cancers. (**B**–**D**): Mutational panel of hotspots for *RET* activating mutations including relevant RET domains and amino acid position in the horizontal axis as well as cumulative number of *RET* mutations depicted in the vertical axis. (**B**): All *RET* mutations, including oncogenic mutations and variants of unknown significance, reported across the combined MSK IMPACT, The Cancer Genome Atlas (TCGA) Pan-Cancer Atlas, and China Pan-Cancer cohorts (approximately 32,000 samples). (**C**): Well-described oncogenic or likely oncogenic *RET* mutations in medullary thyroid cancer (MTC). (**D**): Oncogenic or likely oncogenic *RET* mutations across additional (non-MTC) tumor sites in pooled patients from MSK IMPACT, The Cancer Genome Atlas (TCGA) Pan-Cancer Atlas, and China Pan-Cancer cohorts.

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
