# Peer review of "RET-Altered Cancers—A Tumor-Agnostic Review of Biology, Diagnosis and Targeted Therapy Activity"

_cancers, 2023, doi:10.3390/cancers15164146_

Round 1

Reviewer 1 Report

The review summarizes the clinical and biological aspects of RET fusion in a very comprehensive and well-written way. The paper may be accepted for publication in its current form.

Author Response

Thank you for your feedback and your interest in our review.

Reviewer 2 Report

Review comments for “Biology, Diagnosis and Activity of Targeted Therapy in RET- 2

Altered Cancers – A Tumor-Agnostic Review”

Summary:

Recent advances in cancer treatment involve molecular profiling and inclusive trial designs. Basket trials now enroll patients based on specific oncogenic drivers, like RET alterations, regardless of tumor type. Selpercatinib, a selective tyrosine kinase inhibitor targeting RET, received regulatory approval for multiple tissue-agnostic indications in oncology. The article discusses the biology of RET alterations and strategies to target these drivers using selpercatinib and pralsetinib trials' data.

Comment:

This is a great review. The authors not only introduced latest research and clinical trial progress about RET in different cancer types, but also showed the Standard methods for RET detection now. In the meantime, the authors provided the future directions for the future research in the field which may contribute to exploring the potential therapies in clinical.

Author Response

Thank you for your feedback and your interest. We appreciate your input.

Reviewer 3 Report

This is a very-well written, thorough, and scientifically sound review which addresses all relevant aspects of the chosen topic.

I have some minor suggestions to the authors.

1) In Figure 1, please explain what is meant by the arrow pointing upwards to the lipid raft on the right-hand site.

2) In line 201, you cite a paper regarding the prognostic impact of RET fusions in advanced colorectal cancer. Although I did not find this information in the original publication, I believe that in this cohort the many MSI-H patients did not recieve immune checkpoint inhibitors, as the paper predates the respective EMA approval and it is an Italian cohort. There is therefore reason to believe that the prognostic impact of RET fusions is actually not applicable to the current patient reality, where MSI-H patients (probably including cases with RET rearrangements) have a much more favorable prognosis. Please either comment on this issue or leave out the section.

3) In section 5, you describe outcome data of RET-rearranged NSCLC to immunecheckpoint inhibitor monotherapy (IO). I aggree that the limited data does indicate very low activity of IO in these patients and data on smoking, TMB and PD-L1 support this notion. Please comment on chemo-IO combination therapy. I believe, that this would actually be the only reasonable alternative treatment to RET inhibitors in the firstline palliative setting (e.g. IMpower150 regimen). The addition of chemo to IO led to increased sensitivy of populations otherwise poorly responding to IO monotherapy including PD-L1 low patients and even EGFR+ NSCLC (ORIENT31). This treatment has the potential for long-term remission (>5 years), maybe cure, which has not been described with targeted therapy. On the other hand, recent trials from the EGFR setting (Checkmate722,  Keynote 789) may not support chemo-IO in non-smoking, driver-positive NSCLC. I believe the only study directly addressing the comparison of firstline chemo-IO to RET inhibitors in RET+ NSCLC is Popat et al. (your reference 154), a retrospective analysis which you mention later. A prospective trial is still ongoing NCT04222972. I believe, this issue deserves a thorough comment.

4) Please indicate that when you speak of "approval" you always refer to the FDA only.

5) From a clinical perspective, I would be interested whether resistance mechanisms to selpercatinib/pralsetinib have succesfully be harnessed therapeutically. Please consider https://doi.org/10.1016/j.cllc.2022.08.010  https://doi.org/10.1158/1078-0432.CCR-20-2278

6) In section 9, you suggest universal screening for RET alterations. I would argue that also a two-step process would be reasonable and potentially more cost-effective, where only driver-negative tumors are assessed for rare fusion events (especially in pancreatic or gastrointestinal neoplasia, where testing BRAF and RAS only covers the majority of driver mutation).

Author Response

1) In Figure 1, please explain what is meant by the arrow pointing upwards to the lipid raft on the right-hand site.

SRC requires lipid rafts integrity for activation of downstream signaling.
We added the following text in Figure 1 to reflect this precision: "These phosphorylated tyrosine residues serve as docking sites for adaptor proteins involved in the propagation of RET signaling, including MAPK, PI3K/AKT, PLCγ and SRC (which requires association with lipid rafts for activation)".

2) In line 201, you cite a paper regarding the prognostic impact of RET fusions in advanced colorectal cancer. Although I did not find this information in the original publication, I believe that in this cohort the many MSI-H patients did not receive immune checkpoint inhibitors, as the paper predates the respective EMA approval and it is an Italian cohort. There is therefore reason to believe that the prognostic impact of RET fusions is actually not applicable to the current patient reality, where MSI-H patients (probably including cases with RET rearrangements) have a much more favorable prognosis. Please either comment on this issue or leave out the section.

Correct, we updated the language accordingly: "This series should be interpreted with caution as study findings predate immune checkpoint inhibitor (ICI) US FDA approval for patients with MSI-high tumors; ICI use in this subgroup is active and may impact survival outcomes."

3) In section 5, you describe outcome data of RET-rearranged NSCLC to immune checkpoint inhibitor monotherapy (IO). I aggree that the limited data does indicate very low activity of IO in these patients and data on smoking, TMB and PD-L1 support this notion. Please comment on chemo-IO combination therapy. I believe, that this would actually be the only reasonable alternative treatment to RET inhibitors in the first-line palliative setting (e.g. IMpower150 regimen). The addition of chemo to IO led to increased sensitivity of populations otherwise poorly responding to IO monotherapy including PD-L1 low patients and even EGFR+ NSCLC (ORIENT31). This treatment has the potential for long-term remission (>5 years), maybe cure, which has not been described with targeted therapy. On the other hand, recent trials from the EGFR setting (Checkmate722,  KEYNOTE-789) may not support chemo-IO in non-smoking, driver-positive NSCLC. I believe the only study directly addressing the comparison of firstline chemo-IO to RET inhibitors in RET+ NSCLC is Popat et al. (your reference 154), a retrospective analysis which you mention later. A prospective trial is still ongoing NCT04222972. I believe, this issue deserves a thorough comment.

We will comment on chemo-immunotherapy in a new paragraph which was added to the manuscript in section 5: "Little data is available on the efficacy of chemoimmunotherapy in patients with RET-altered cancers. In patients with RET fusion-positive NSCLCs, the combination of platinum-based chemotherapy and pembrolizumab in the first-line setting was associated with inferior OS and PFS when compared to pralsetinib in a retrospective analysis. Prospective data will be generated by the first line AcceleRET and LIBRETTO-431 studies that randomize patients with RET fusion-positive lung cancers to selective RET inhibitor therapy or chemotherapy +/- immunotherapy."

4) Please indicate that when you speak of "approval" you always refer to the FDA only.

We now specified that "approval" refers to FDA approval in lines 22, 30, 287, 435, 572, 667.

5) From a clinical perspective, I would be interested whether resistance mechanisms to selpercatinib/pralsetinib have succesfully be harnessed therapeutically. Please consider https://doi.org/10.1016/j.cllc.2022.08.010, https://doi.org/10.1158/1078-0432.CCR-20-2278

We added the following sentence to incorporate therapeutic considerations in patients with acquired MET amplifications: "Proof of concept clinical reports have demonstrated the activity of MET inhibitors in the setting of acquired MET amplification in patients with RET fusion-positive NSCLCs. The combination of crizotinib and selpercatinib led to responses in selpercatinib-refractory patients with acquired MET amplification. In a case report, sequential use of crizotinib and capmatinib led to a response in a patient with KIF5B-RET fusion-positive NSCLC with acquired MET amplification who had initially progressed on pralsetinib."

6) In section 9, you suggest universal screening for RET alterations. I would argue that also a two-step process would be reasonable and potentially more cost-effective, where only driver-negative tumors are assessed for rare fusion events (especially in pancreatic or gastrointestinal neoplasia, where testing BRAF and RAS only covers the majority of driver mutation).

We also agree on this comment and added the following sentence (lines 690-692) for clarification purposes: "Alternatively, a two-step approach would be reasonable, with screening for RET alterations limited to patients without oncogenic drivers on initial testing."